# *Piscirickettsia salmonis*-Triggered Extracellular Traps Formation as an Innate Immune Response of Atlantic Salmon-Derived Polymorphonuclear Neutrophils

**DOI:** 10.3390/biology10030206

**Published:** 2021-03-09

**Authors:** Pablo Alarcon, Gabriel Espinosa, Catalina Millan, Julia Saravia, Vania Quinteros, Ricardo Enriquez, Claudio Henriquez, Luis Vargas-Chacoff, Rafael A. Burgos, Anja Taubert, Carlos Hermosilla, Francisco J. Morera

**Affiliations:** 1Laboratory of Molecular Pharmacology, Instituto de Farmacologia and Morfofisiologia, Facultad de Ciencias Veterinarias, Universidad Austral de Chile, Valdivia 5090000, Chile; rburgos1@uach.cl; 2Applied Biochemistry Laboratory, Instituto de Farmacologia y Morfofisiologia, Facultad de Ciencias Veterinarias, Universidad Austral de Chile, Valdivia 5090000, Chile; essgabo@gmail.com (G.E.); catalina.millanb@gmail.com (C.M.); 3School of Medicine, Universidad Austral de Chile, Valdivia 5090000, Chile; 4Fish Physiology Laboratory, Instituto de Ciencias Marinas y Limnologicas, Facultad de Ciencias, Universidad Austral de Chile, Valdivia 5090000, Chile; saravia.julia@gmail.com (J.S.); luis.vargas@uach.cl (L.V.-C.); 5Centro Fondap de Investigación de Altas Latitudes (IDEAL), Universidad Austral de Chile, Valdivia 5090000, Chile; 6Doctorado en Ciencias de la Acuicultura, Universidad Austral, Puerto Montt 5480000, Chile; 7Instituto de Patologia Animal, Facultad de Ciencias Veterinarias, Universidad Austral de Chile, Valdivia 5090000, Chile; vaniaquinteros@gmail.com (V.Q.); renrique@uach.cl (R.E.); 8Instituto de Farmacologia and Morfofisiologia, Facultad de Ciencias Veterinarias, Universidad Austral de Chile, Valdivia 5090000, Chile; claudio.henriquez@uach.cl; 9Integrative Biology Group, Universidad Austral de Chile, Valdivia 5090000, Chile; 10Biomedical Research Center Seltersberg, Institute of Parasitology, Justus Liebig University Giessen, 35390 Giessen, Germany; anja.taubert@vetmed.uni-giessen.de (A.T.); Carlos.R.Hermosilla@vetmed.uni-giessen.de (C.H.)

**Keywords:** extracellular traps, Atlantic salmon, *Piscirickettsia salmonis*, innate immunity, *Salmo salar*

## Abstract

**Simple Summary:**

Within innate immunity, polymorphonuclear neutrophils (PMN) are the most abundant leukocyte population. Alongside PMN, monocytes, eosinophils, and basophils are also known to exist. All of them can release extracellular traps (ETs), a complex web-like structure composed of chromatin decorated with nuclear histones, granular enzymes, peptides, and proteins, to firmly entrap invasive pathogens, thereby slowing dissemination and helping to develop proper immune responses against bacteria, fungi, viruses, and parasites. Here, we showed for the first time that Atlantic salmon-derived PMN released ETs-like structures in vitro, in response to highly pathogenic facultative intracellular rickettsial bacteria *Piscirickettsia salmonis*. The release of ET-like structures from PMN could be a new alternative to improve farmed salmon’s defense against pathogens.

**Abstract:**

Extracellular traps (ETs) are webs of DNA, citrullinated histones, anti-microbial peptides, and proteins that were not previously reported in Atlantic salmon (*Salmo salar*). ETs are mainly released from polymorphonuclear neutrophils (PMN) and are considered a novel PMN-derived effector mechanism against different invasive pathogens. Here, we showed that Atlantic salmon-derived PMN released ETs-like structures in vitro in response to highly pathogenic facultative intracellular rickettsial bacteria *Piscirickettsia salmonis*. PMN were isolated from pre-smolt Atlantic salmon and stimulated in vitro with oleic acid and *P. salmonis.* Extracellular DNA was measured using the PicoGreen™ dye, while immunofluorescence image analysis was used to confirm the classical components of salmonid-extruded ETs. Future studies are required to better understand the role of Atlantic salmon-derived ETs orchestrating innate/adaptive immunity and the knowledge on regulation pathways involved in this cell death process. Thus, comprehension of salmonid-derived ETs against *P. salmonis* might represent novel alternative strategies to improve host innate defense mechanisms of farmed salmon against closely related rickettsial bacteria, as a complement to disease prevention and control strategies.

## 1. Introduction

Multicellular organisms developed complex host immune responses to defend themselves against invasive pathogens. Host immune responses are classically divided into innate immune responses, which react early, fast, and non-specifically upon first encounters with pathogens, and adaptive (acquired) immune responses, which are slower to develop but are specific and result in immunological memory [1,2]. These two types of immune responses are found in all classes of vertebrate species, including fish [3].

Within host innate immunity, polymorphonuclear neutrophils (PMN) are the most abundant leukocyte population. Alongside PMN, also monocytes, eosinophils, and basophils are known to exist. Regardless of functional differences between these leukocytes, all of them can release extracellular traps (ETs), a complex web-like structure composed of chromatin decorated with nuclear histones, granular enzymes, peptides, and proteins, to firmly entrap invasive pathogens, thereby slowing dissemination and helping to develop proper immune responses against bacteria, fungi, viruses, and parasites [4,5,6,7]. ETs are the result of the process termed ETosis, a controlled cellular death process resulting in the release of extracellular chromatin, either from nuclei or mitochondrial origin, decorated with anti-microbial global histones (H1, H2A/H2B, H3, H4), peptides and enzymes, capable of efficiently entrapping and killing microorganisms in the extracellular space [8,9,10].

As stated before, major structural components of PMN-derived ETs are DNA, histones, and proteins from granules [neutrophil elastase (NE), myeloperoxidase (MPO), cathepsin G, lactoferrin, pentraxin, and gelatinase)] [5,11]. ETosis is induced by several chemical compounds such as the protein kinase C activator, phorbol myristate acetate (PMA), oleic acid (OA), hydrogen peroxide (H_2_O_2_) from the respiratory burst, lipopolysaccharide (LPS), or bacteria [6,12,13,14]. This cellular process is an essential part of the mammalian inflammatory response but is also related to pathogenicity in some diseases, such as thrombosis, mastitis, arthritis, preeclampsia, appendicitis, cancer metastasis, and reproductive disorders [15,16,17,18,19].

Interestingly, it is well documented that ETosis occurs not only in vertebrates but also in invertebrates species [4,20]. Consistently, ETs were also described in PMN-enriched suspensions isolated from the head kidney from teleost fish such as zebrafish (*Danio rerio*) [21] and fathead minnow (*Pimephales promelas*) [22]. Contrary to mammalian-derived ETosis published data, only a few fish-derived ETs are described so far, including common carp (*Cyprinus carpio*) [23,24], turbot (*Scophthalmus maximus*) [25], and sole (*Cynoglossus semilaevis*). Moreover, there is one recent report on ETs release from salmonid species, i.e., rainbow trout (*Oncorhynchus mykiss*) [26], a salmonid with commercial importance worldwide. To our best knowledge, no published studies exist on Atlantic salmon (*Salmo salar*)-derived ETs, despite its economic importance for many countries with large salmon industries, such as Canada, United Kingdom, Norway, and Chile, among others. The same holds for detailed investigations on Atlantic salmon-mediated ETosis inducers, such as salmon-specific pathogens or components. 

*Piscirickettsia salmonis* is a Gram-negative, non-motile rickettsial bacteria [27,28], which was initially described as an obligate intracellular organism [29]. Only recently, *P. salmonis* was reclassified as a facultative intracellular organism [30,31,32]. More importantly, *P. salmonis* is present in Southern Chilean marine waters affecting salmonid farming centers, causing significant mortalities and economic losses for the local salmon industry [33]. According to the Chilean National Department for Fishes and Agriculture (SERNAPESCA [Spanish acronym]), Atlantic salmon is the most cultured salmonid species in Chile, presenting mortality rates of 46.8%, due to *P. salmonis* infections [34]. Coho salmon (*Oncorhynchus kisutch*) as well as rainbow trouts (*O. mykiss*) showed high mortality rates associated with piscirickettsiosis, reaching 14.3% and 46.6%, respectively [34]. *P. salmonis* was the first rickettsial-like Gram-negative bacterium known as a fish pathogen [35]. Clinical presentation of piscirickettsiosis includes a systemic and chronic infection that affects marine salmon cultures without age discrimination. There is a wide range of external clinical signs described in *P. salmonis*-infected salmonids (for a more detailed review, please refer to [33]). Additionally, this pathogen cause endogenous infections of the liver, spleen, serosanguinous ascites, and swollen kidneys (nephritis) [33,36,37].

The present study aims to assess if salmonid-derived ET-like structures are functionally presented in isolated PMN from Atlantic salmon and whether these ET-like structures are released in response to *P. salmonis*, in order to better understand the possible role of ancient ETs reactions against other closely related bacteria of Atlantic salmon.

## 2. Materials and Methods

### 2.1. Fish and Sampling

Atlantic salmon (*S. salar*) were maintained in indoor tanks supplied with recirculating freshwater at ambient temperatures in the Fish Vivarium of the Laboratory of Salmon Clinical Trials (SCT-Lab), at the Institute of Animal Pathology, Faculty of Veterinary Sciences of the Universidad Austral de Chile, Valdivia, Chile. Fish were fed with commercial pellets. Sampling was conducted according to the principles, the procedures were approved by the Institutional Animal Care Committee of the Universidad Austral de Chile (Code no. 318/2018), and the protocols were established in the SCT-Lab (Code no. R.N.A. 140008). For sampling, fish were quickly dip-netted out of the tanks and anesthetized with 30 mg/L benzocaine (BZ-20; Veterquimica S.A., Santiago, Chile). 

### 2.2. Bacteria Culture

A field-collected isolate of the *P. salmonis* strain LF-89 from the Fish Pathology Laboratory Collection at Universidad Austral de Chile was grown under standard conditions [32] on AUSTRAL-TSFe agar plates at 18 °C, for at least 14 days, until a film of bacteria was formed on the agar plate. Later, 0.50 ± 0.04 cm^2^ area of the bacteria was re-suspended in 1 mL of sterile HBSS with 0.9 mM of calcium and slightly vortexed for 15 s. Then, *P. salmonis* microbes were warmed at 37 °C for 5 min before exposing 10 and 50 μL of this *P. salmonis* containing solution to Atlantic salmon-derived PMN.

### 2.3. Obtaining a PMN-Enriched Suspension from Atlantic Salmon

The isolation of salmon-derived PMN from head kidney samples was conducted according to previously published reports on fish ETs [21,22]. Briefly, one-third of the kidney, corresponding to the hematopoietic tissue, was placed in sterile plates (Gibco) containing 5 mL of sterile PBS (10 mM sodium phosphate, 2.7 mM KCl, and 137 mM NaCl, pH 7.4). Then, the tissue was mechanically disaggregated to release cells from the extracellular matrix. Cellular debris was eliminated with a 100-µm cell strainer (Falcon, Thermo Fisher). The cells were placed on a discontinuous density gradient (Percoll, GE Healthcare, Chicago, IL, USA), with 4 mL of Percoll 1.08 g/mL (diluted with 1.5 M NaCl) in the bottom of a 15-mL plastic tube (Falcon) with 4 mL of sterile Percoll 1.06 g/mL (diluted with 1.5 M NaCl) above. After centrifugation (400× *g,* 70 min, 4 °C), the upper layer contained mononuclear cells, and the lower layer contained granulocytes. Both layers were carefully aspirated for further processing. The viability of the obtained PMN was 85% ± 5 using the trypan blue exclusion test (0.4%; Sigma-Aldrich, San Luis, MO, USA). The purity of the obtained PMN was analyzed by flow cytometry (BD FACSCanto II; Becton Dickinson, Franklin Lakes, NJ, USA), using a forward-scatter versus side-scatter dot plot, to determine the relative size and granularity of PMN. A characteristic forward-scatter versus side-scatter dot plot is shown in Figure 1A. Cells were subsequently prepared for further bioassays.

### 2.4. Total Reactive Oxygen Species (ROS) Production

To assess the effect of *P. salmonis* on respiratory burst as a control of salmon-derived PMN functionality, we used the luminol-chemiluminescence assay, as described previously [38,39]. In brief, freshly isolated salmon-derived PMN (1 × 10^6^ cells/wells) were seeded in 96-well plate and pre-warmed at 37 °C for 10 min. Later, 100 μM luminol was added and after that incubated for 5 min at 37 °C. After 100 s, 10 and 50 µL of *P. salmonis* or 100 nM of platelet activating factor (PAF) or vehicle stimulation was added. ROS production was recorded for 1500 s using a Luminoskan Ascent^®^ microplate reader (Thermo Scientific, Waltham, MA, USA), at 37 °C. The luminescence was measured in relative light units (RLU) and expressed as the peak maximum.

### 2.5. Quantification of P. salmonis-Triggered ETs

*P. salmonis*-triggered ETs were quantified by measuring the amount of free extracellular DNA, which was stained by the cell-impermeant fluorescent dye PicoGreen^®^ (Invitrogen, Carlsbad, CA, USA), according to Muñoz-Caro et al. [40] and Alarcón et al. [12]. A total of 1 × 10^6^ PMN/tube were incubated at 37 °C and then stimulated with different volumes of *P. salmonis* (10 and 50 µL), or 300 µM of oleic acid (OA) or vehicle (HBSS/calcium), for 60 min. Micrococcal nucleases were added (5 U/well, New England Biolabs, Ipswich, MA, USA) to each condition and incubated for 15 min at 37 °C. Samples were centrifuged (300× *g*, 5 min), and 100 µL of the supernatants were transferred into a 96-well plate, and PicoGreen^®^ (50 mL/well, diluted 1:200 in 10 mM Tris/1 mM EDTA) was added. ET formation was determined by spectrofluorometric analysis (484 nm excitation/520 nm emission), using an automated reader (Varioskan Flash^®^; ThermoFisher Scientific, Waltham, MA, USA), as previously described [40].

### 2.6. Visualization of P. salmonis-Triggered ETs 

In total, 2 × 10^5^ PMN were suspended in 200 µL of sterile HBSS with 0.9 mM calcium after stimulation, and the cells were deposited on coverslides previously pre-coated with 0.003% poly-l-lysine (Sigma-Aldrich, San Luis, MO, USA), for 30 min at 37 °C. Thereafter, the cells were fixed by adding 200 µL of 4% paraformaldehyde (Merck) for 30 min at room temperature (RT). Subsequently, the cells were washed twice with sterile HBSS and blocked with 1% bovine serum albumin (BSA for 2 h; Sigma-Aldrich) in water. To visualize the nuclei, the coverslides were stained with PicoGreen^®^ (S34861, Thermofisher, Waltham, MA, USA), 1:200 in sterile HBSS with calcium, for 30 min at RT in darkness, and then again washed three times with sterile HBSS and carefully mounted with Mounting^®^ anti-fading buffer (DAKO, Agilent, Santa Clara, CA, USA).). Finally, fluorescence images were acquired with a Fluoview 1000^®^ confocal microscope (Olympus, Miami, FL, USA).

### 2.7. Statistics

Data are expressed as mean ± SEM, and for comparison between treatments, one-way analysis of variance (ANOVA) was performed, and Fisher’s LSD multiple comparison test was applied, using a significance level of 5%. When assumptions of normality or homogeneity of variance were not met according to the Shapiro–Wilks or Brown–Forsythe test, respectively, Kruskal–Wallis ANOVA and Dunn’s multiple comparison tests were used. All statistical analyses were performed using the GraphPad^®^ Prism v7.0 (GraphPad Software, La Jolla, CA, USA). A *p*-value < 0.05 was considered significant.

## 3. Results and Discussion

### 3.1. ROS Production in P. salmonis-Exposed PMN from Atlantic Salmon 

Fish immune system shows similar features to those of vertebrates [1,2]. Within teleost fish innate immune system are leukocytes such as monocytes, non-specific cytotoxic cells (or NK cells), PMN, eosinophils, and basophils [41,42]. As for other vertebrate species, PMN are the most abundant leukocyte population in the teleost fish blood [21]. PMN isolation from the head kidney of Atlantic salmon showed similar characteristics in the side vs. scatter plot FACS analysis, fitting well to a previous report on fish-derived PMN (Figure 1A) [43].

ROS production is a fundamental response in leukocytes, allowing defense against invasive fish pathogens [44]. ROS is produced by multicomplex enzymatic protein NADPH oxidase (NOX), which catalyzes superoxide anion production [45]. In Nile tilapia (*Oreochromis niloticus*), it was described that diazinon (an organophosphorus pesticide) increased ROS production of fish leukocytes extracted from blood obtained via cardiac puncture [46]. Additionally, prolactin induced ROS production via the PKC signaling pathway in leukocytes extracted from bony fish gilthead seabream. Classical ROS production inductors in human and mammalian PMN, such as zymosan and PMA, induced NOX activation in goldfish kidney-isolated PMN [47]. Samai et al. [43] evaluated fish-derived ROS production of leukocytes isolated from head kidney, spleen, and blood using PMA as an ROS inducer. Former authors found that isolated PMN from head kidney responded with strong ROS production after PMA stimulation [43].

We found that after purification, PMN showed a high basal ROS production level (Figure 1B), characteristically seen in non-stimulated PMN after isolation. Similar PMN features were reported by Samai et al. [43]. Interestingly, PMN treated with 100 nM PAF increased ROS production significantly (please refer to Figure 1C), meaning that salmon-derived PMN were still functional after isolation. PAF is a classical stimulus for strong ROS production in human-, bovine- [48], and canine PMN [49], among other species. Indeed, it was demonstrated that PAF possesses anti-thrombotic effects in Atlantic salmon (*S. salar*) in vivo [50]. Nonetheless, salmon-derived PMN increased ROS production after being treated with 50 µL of the *P. salmonis* culture but not with 10 µL of *P. salmonis* (Figure 1D–F). In summary, we demonstrated for the first time that both highly pathogenic *P. salmonis* and PAF, induced a transient peak of ROS production in salmonid-isolated PMN.

### 3.2. Atlantic Salmon-Derived PMN Release ETs after Exposure to P. salmonis 

Another effector mechanism displayed by PMN is the formation of extracellular traps (ETs) [5]. This response was described in different leukocytes types, such as neutrophils [5,6,7], eosinophils [51], basophils [52], monocytes [53,54], and plasmacytoid dendritic cells [55]. In line, ET formation was described in macrophages [56] and mast cells [57]. Several stimuli could trigger ETosis, such as bacteria, fungi, parasites, viruses, and pro-inflammatory stimuli (for more detailed review [58]). To test if isolated *S. salar*-derived PMN could release ETs in response to the presence of a common salmon aquaculture rickettsial bacterial pathogen, we incubated isolated PMN with *P. salmonis* for 60 min. We observed an increase of cell free DNA (cf-DNA) after 10 and 50 µL of *P. salmonis* stimulation (Figure 2A). In addition, we used OA as positive control, which was previously described as a potent NETs inducer in bovine PMN [12]. To corroborate *P. salmonis*-induced ETs in Atlantic salmon-isolated PMN, we used a fluorescent dye for extracellular DNA staining (PicoGreen^®^), to visualize some thin extracellular web-like structures after either *P. salmonis*- or OA stimulation (white arrows in Figure 2B). Zhao et al. [59] demonstrated that PMA, as well as fish-specific pathogens, namely *Edwardsiella tarda, Pseudomonas fluorescens,* and *Vibrio harveyi,* also resulted in ETosis in head kidney PMN isolated from tongue sole (*Cynoglossus semilaevis*).

Interestingly, our results showed that the lower volume of the bacterial suspension failed to induce a measurable increase in ROS production but was able to produce a significant increase in the release of ETs from the isolated leukocytes. Currently, in mammals, the relation between ROS production and ETs liberation is controversial. A large body of evidence connects both mechanisms and supports that the release of ETs requires ROS production, either via the action of NADPH or through mitochondrial metabolism [60,61]. 

Recognition of microbial ligands, or pathogen-associated molecular patterns (PAMPs) through pattern-recognition receptors (PRRs), stimulates host innate immune cells in order to upregulate the expression of cytokines, chemokines, and proteins that directly target microbes [62]. It was described that NETs were absent in mice lacking MyD88, a key component of the TLR-dependent signaling pathway, indicating that NETs are dependent on TLR signaling [63]. 

*P. salmonis* is an aggressive fish pathogen that threatens salmon production sustainability in Chile [64]. Interestingly, *P. salmonis* can escape into the macrophage cytoplasm and remain at least partly enclosed within a vacuole membrane [65]. Consistently, *P. salmonis* is known to hamper innate immune reactions by inducing apoptosis in macrophages and monocyte-like cells [66]. Even though precise molecular invasion mechanisms used by *P. salmonis* to infect host fish cells are not entirely understood [67], there is experimental evidence of *P. salmonis*-associated exotoxin secretion to achieve host cell invasion [68]. Irrespective of exotoxin-dependent invasion, it is further known that certain bacterial exotoxins can effectively modulate ET formation, and even other microbial exotoxins might contribute to ET formation enhancement (reviewed by [69]).

The process of ETosis avoids the spread of invasive pathogens within the host and hampers host cell invasion by immobilizing pathogens, as recently demonstrated for obligate intracellular apicomplexan parasites [70]. However, a persistent piscirickettiosis could also act as a double-edged sword for this specific PMN response, due to exacerbation of ET-release like proteolytic and antibacterial proteins, which decorate the thin networks of extruded DNA, pro-inflammatory peptides, enzymes, and pan histones (H1, H2A/H2B, H3, H4), which can damage tissues or organs neighboring the infection site [71,72]. As such, apicomplexan parasite *Besnoitia besnoiti*-induced ETs resulted in H2A-dependent damage of the primary endothelium [73].

The release of *P. salmonis*-mediated ET-like structures in Atlantic salmon strongly suggests that this ancient and well-conserved host innate effector mechanism could be acting against other related infectious pathogens. Further studies are needed to verify whether this phenomenon also occurs in vivo and to better understand its possible contribution to the host defense against relevant aquaculture pathogens. However, it opens a new possibility for investigating in more detail the molecular components, signaling pathways, and receptors in *P. salmonis*-mediated ETosis.

Aquaculture farming in Chile experienced a swift growth, mainly due to favorable geographical and environmental conditions. Currently, Chile is one of the ten major aquaculture producer countries and the second largest Atlantic salmon producer in the world, with an annual average volume of close to 800,000 tons between 2014 and 2018 [34]. Nevertheless, this rapid growth, together with confinement systems and the presence of native marine life, increases the risk of infectious disease outbreaks. Even in years without epidemics, contagious diseases are always among the leading causes of death and economic losses in the Chilean salmon industry [34]. The release of ET-like structures from PMN could be modulated through food supplements, probiotics, or pharmacological compounds [74]. These possibilities open new alternatives to improve farmed salmon defense against pathogens, for which there are no vaccines available yet. They might contribute to reducing the use of antibiotics, thereby diminishing antibiotic multiresistance development not only in Chilean salmonid aquaculture but also within the fragile marine ecosystem.

## Figures and Tables

**Figure 1 biology-10-00206-f001:**
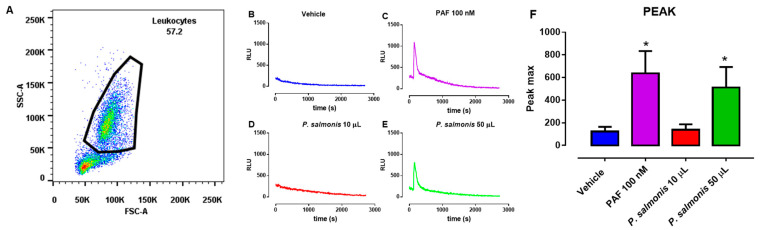
ROS production of PMN from Atlantic salmon (*Salmo salar*)*. (***A**) Dot graph (scatter v/s forward graph) of PMN population isolated from anterior kidney of Atlantic salmon (*Salmo salar*). Representative image of ROS production of PMN treated with (**B**) vehicle (HBSS with Calcium), (**C**) PAF 100 nM (positive control), and (**D**) 10 µL of *Piscirickettsia salmonis* (*P. salmonis*), or (**E**) 50 µL of *P. salmonis*. (**F**) Bar graph of peak maximum ROS production of PMN treated with *P. salmonis*, PAF, or vehicle. *n* = 5. * *p* < 0.05 compared to vehicle.

**Figure 2 biology-10-00206-f002:**
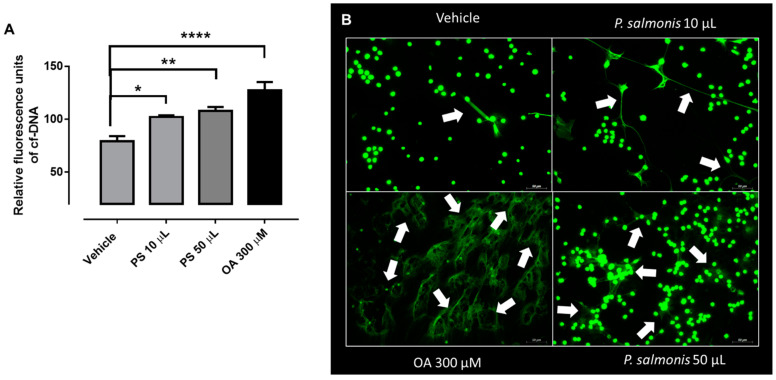
*Piscirickettsia salmonis* induced ETosis in PMN from Atlantic salmon (*Salmo salar*)*. (***A**) Quantification of extracellular DNA (cf-DNA: cell free DNA) of Atlantic salmon (*Salmo salar*) PMN, treated with 10 µL and 50 µL of *P. salmonis* (PS), 300 µM oleic acid (OA) or vehicle for 30 min. *n* = 5. * *p* < 0.05; ** *p* < 0.01; **** *p* < 0.0001. (**B**) Representative images of *Salmo salar*-derived PMN treated with vehicle (HBSS + Calcium), 10 µL, and 50 µL of *P. salmonis*. As a positive control of ETosis, we add 300 µM OA for 60 min. Green represents DNA structures. White arrow indicates ET-like structures. Scale bar is 50 µm.

## Data Availability

Not applicable.

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
