# Peer review of "Piscirickettsia salmonis-Triggered Extracellular Traps Formation as an Innate Immune Response of Atlantic Salmon-Derived Polymorphonuclear Neutrophils"

_biology, 2021, doi:10.3390/biology10030206_

Round 1
Reviewer 1 Report
The authors describe experimental ROS production and ET stimulation of isolated PMNs from Atlantic salmon. The experiments appear to be conducted well and statistical analysis seems sound. The presentation of their data is clear and the discussion is mostly conclusive. In summary, the manuscript provides a valuable contribution to the field of non-mammalian ET-formation, a topic that is still often disregarded in ETosis research, despite its importance for our understanding of the phylogeny and evolution of this defence mechanism as well as for the development of veterinary or agricultural applications.
Suggested changes:
Parts of the Methods section are difficult to follow, as a lot of cross-referencing is needed. This could be improved by adding a paragraph describing cell culture conditions and stimulation protocols separately from the detection methods.
The last paragraph of section 3.1 should be reorganised so that the PAF-results are mentioned before the P. salmonis results.
In the last paragraph of the Discussion section an application for aquaculture is suggested. However it is unclear to me what exactly such an application would involve.
Adding a list of abbreviation would be beneficial.
Minor points:
Introduction, 2nd paragraph:
“ETs are also described as ETosis” should be rephrased, as ETs are the result of the process termed ETosis.
Introduction, 4th paragraph: omit “first”
Methods, 2.4 ROS Production (and following):
- “adding 10 and 50 μL of P. salmonis” please give the concentrations in cells/ml
- “PAF” write out full word
Methods, 2.5 Quantification of ETs:
“the cell-permeant fluorescent dye PicoGreen” I might be mistaken but I think it should be cell-impermeant
Results and Discussion, 3.1 last paragraph: Replace “finally” by “in summary”
Results and Discussion, 3.2:
- “This response has been described in different leukocytes types such as macro-phages [51], eosinophils [52], basophils [53], monocytes [54,55], mast cells [56], and plasmacytoid dendritic cells [57].” ETs have been described mainly in neutrophils which aren't mentioned here.
- “we visualized using a fluorescent dye ” this phrase lacks a noun
- “head kidney-isolation PMN from tongue” I’m not sure if this is correct English grammar
- 4th paragraph “P. salmonis can escape into the macrophage cytoplasm and remaining at least partly enclosed” remain (omit “ing”)
- 5th paragraph “ETosis process avoids the spread of invasive pathogens” Add ‘The’
- “pan histones (H1, H2A/H2B, H3, H4) can induce damage tissues/organs” change to: “damage to tissues/organs”
Figure 2, legend:
“PMN treated with 10 μL and 50 μL of P. salmonis (PS), 300 μL OA” give concentration in mM for OA
“White arrow indicates structures like-ET” change to “White arrows indicate ET-like structures“
See also comments in attached PDF

Author Response
Answers to Reviewers` comments
(Reviewers’ comments are italicized below)
We thank the reviewer by his very useful comments that have greatly improved our paper.
Reviewer 1: Comments and Suggestions for Authors
The authors describe experimental ROS production and ET stimulation of isolated PMNs from Atlantic salmon. The experiments appear to be conducted well and statistical analysis seems sound. The presentation of their data is clear and the discussion is mostly conclusive. In summary, the manuscript provides a valuable contribution to the field of non-mammalian ET-formation, a topic that is still often disregarded in ETosis research, despite its importance for our understanding of the phylogeny and evolution of this defence mechanism as well as for the development of veterinary or agricultural applications.
Suggested changes:
- Parts of the Methods section are difficult to follow, as a lot of cross-referencing is needed. This could be improved by adding a paragraph describing cell culture conditions and stimulation protocols separately from the detection methods.
Answer: Thank you so much for your constructive comments concerning the section ‘Material and methods’. Nonetheless, we partially agree on this observation. As you will read from the revised manuscript, we have already added adequate subheadings to each topic related to this investigation. There are sufficient data provided in each subheading section to gain sufficient on local salmon facilities at the UACH (Chile), the PMN isolation protocols, FACS-analyses, bacterial culture, etcetera. Actually, our serious intention was not to complicate sentences by adding more and more information and instead focusing on essential steps. In the past, we have published various manuscripts on pathogen-mediated ETs, -NETs, and -METs by following this description, and to our humble opinion, it would complicate the reading by describing cell culture and stimulation protocols separately from the detection methods.
- The last paragraph of section 3.1 should be reorganised so that the PAF-results are mentioned before the P. salmonis results.
Answer: We reorganized the paragraph as the reviewer has suggested.
- In the last paragraph of the Discussion section an application for aquaculture is suggested. However it is unclear to me what exactly such an application would involve.
Answer: We added the following sentence in the last paragraph of the “Results and Discussion” section to give an idea about possible applications: “The release of ET-like structures from PMN can be modulated through food supplements, probiotics, or pharmacological compounds [75]. These possibilities open new alternatives to improve farmed salmon defense against pathogens for which there are no vaccines available yet”.
- Adding a list of abbreviation would be beneficial.
Answer: We added a list of abbreviation after keywords. Thank you for this suggestion.
Minor points:
Introduction, 2nd paragraph:
“ETs are also described as ETosis” should be rephrased, as ETs are the result of the process termed ETosis.
Answer: Done.
Introduction, 4th paragraph: omit “first”
Answer: Done.
Methods, 2.4 ROS Production (and following):
- “adding 10 and 50 μL of P. salmonis” please give the concentrations in cells/ml
Answer: Thanks for your comment. When carrying out this work, one of the limitations was to know the exact amount of the bacteria that we use because this type of bacteria grows better in the form of biofilm than in solution. Therefore, it did not allow us to know the number of bacteria that we use in the assays described. Although we tried to quantify them using a kit for flow cytometry, an unforeseen event in the equipment and the current pandemic made it impossible for us to do so. This does not imply that in the near future, we will be able to do it and that we can use this short communication as a reference for future publications in this area. That is why we established an area of the biofilm dissolved in the 1mL of HBSS solution, and we used 10 and 50 uL from this mix to stimulate. Now, we explained it in Materials and methods 2.2. Bacteria culture with the sentence: “Later on, 0.50 ± 0.04 cm2 area of the bacteria was re-suspended in 1 mL of sterile HBSS with 0.9 mM of calcium and slightly vortex for 15 s. Then, P. salmonis microbes were warmed at 37 °C for 5 min before exposure 10 and 50 μL of this P. salmonis containing solution to Atlantic salmon-derived PMN”.
- “PAF” write out full word
Answer: Done.
Methods, 2.5 Quantification of ETs:
“the cell-permeant fluorescent dye PicoGreen” I might be mistaken but I think it should be cell-impermeant
Answer: This mistake was corrected. Thank you.
Results and Discussion, 3.1 last paragraph: Replace “finally” by “in summary”
Answer: Done.
Results and Discussion, 3.2:
- “This response has been described in different leukocytes types such as macro-phages [51], eosinophils [52], basophils [53], monocytes [54,55], mast cells [56], and plasmacytoid dendritic cells [57].” ETs have been described mainly in neutrophils which aren't mentioned here.
Answer: This mistake was corrected as follow: “This response has been described in different leukocytes types such as neutrophils [5–7], eosinophils [52], basophils [53], monocytes [54,55], and plasmacytoid dendritic cells [57]. In line, ET formation has been described in macrophages [51] and mast cells [56].” Thank you for this correction.
- “we visualized using a fluorescent dye ” this phrase lacks a noun
Answer: We re-wrote this sentence to: “we used a fluorescent dye for extracellular DNA staining (PicoGreen®) to visualize some thin extracellular web-like structures”.
- “head kidney-isolation PMN from tongue” I’m not sure if this is correct English grammar
Answer: We re-wrote the sentence as follows: “Etosis in head kidney PMN isolated from tongue sole (Cynoglossus semilaevis).”
- 4th paragraph “P. salmonis can escape into the macrophage cytoplasm and remaining at least partly enclosed” remain (omit “ing”)
Answer: Done.
- 5th paragraph “ETosis process avoids the spread of invasive pathogens” Add ‘The’
Answer: Done.
- “pan histones (H1, H2A/H2B, H3, H4) can induce damage tissues/organs” change to: “damage to tissues/organs”
Answer: Done.
Figure 2, legend:
“PMN treated with 10 μL and 50 μL of P. salmonis (PS), 300 μL OA” give concentration in mM for OA
Answer: Done. We corrected the legend in the new versión of manuscripts.
“White arrow indicates structures like-ET” change to “White arrows indicate ET-like structures“
Answer: Done.
Reviewer 2 Report
The manuscript describes the demonstration of extracellular trap formation by head kidney derived Atlantic salmon polymorphonuclear neutrophils (PMN) in vitro following exposure to the fish pathogenic rickettsial bacteria Piscirickettsia salmonis. P. salmonis is a pathogen responsible for significant losses in salmon aquaculture and knowledge of defence mechanisms in salmon may contribute to strategies for improved fish health. The results are clearly presented but the manuscript requires minor revision and text editing before it is acceptable for publication.
Minor revisions
- Plural form of Fish names. Use the singular form for the plural of a fish name, e.g. use salmon not salmons; carp not carps, sole not soles. Also use the singlular form fish (not fishes) when referring to a group of individuals of the same species.
- Section 3.2 Second sentence. Macrophages and mast cells are not leukocytes. This sentence should be modified to avoid describing these two cell types as leukocytes.
- Awkward or long sentences.
- Consider re-writing the final sentence in the second last paragraph in Introduction. "Also, endogenous…immune reactions."
- Section 3.2 First paragraph. Consider "...we used a fluorescent dye...to visualize…"
- Section 3.2 Second paragraph. Sentence starting "Nonetheless,..." is difficult to understand.
- Section 3.2 5th paragraph Consider using "The process of ETosis avoids… and hampers..."
- Repeated text: Introduction (second sentence) and Results and Discussion 3.1 (first sentence) are very similar. Alternative formulation should be used to avoid repetition.
- Formatting.
- Section 2.2 First sentence Degree symbol is underlined
- Section 2.3 Replace comma that is used as decimal point
- Section 3.2 Missing bracket at end of Third sentence
- Spelling.
- Abstract. Use image analysis not imaging analysis.
- Section 2.6 In title, replace de with of
- Section 3.2 Second paragraph final sentence. where not were
- Section 3.2 Fourth paragraph. remain not remaining
- Word missing.
- Introduction. End of second paragraph. "...in the extracellular."
- Section 3.2 5th paragraph "...damage of tissue/organs..."
- Consider using alternative Word.
- Introduction. Second last paragraph. "Inline" or rewrite the sentence.
- Section 3.1 Second paragraph "Former authors.." Consider e.g. "Samai and coworkers…"
- Section 3.2 First paragraph 6th sentence. "Besides" e.g. In addition
Author Response
Answers to Reviewers` comments
(Reviewers’ comments are italicized below)
We thank the reviewer by his very useful comments that have greatly improved our paper.
Reviewer 2: Comments and Suggestions for Authors
The manuscript describes the demonstration of extracellular trap formation by head kidney derived Atlantic salmon polymorphonuclear neutrophils (PMN) in vitro following exposure to the fish pathogenic rickettsial bacteria Piscirickettsia salmonis. P. salmonis is a pathogen responsible for significant losses in salmon aquaculture and knowledge of defence mechanisms in salmon may contribute to strategies for improved fish health. The results are clearly presented but the manuscript requires minor revision and text editing before it is acceptable for publication.
Minor revisions
- Plural form of Fish names. Use the singular form for the plural of a fish name, e.g. use salmon not salmons; carp not carps, sole not soles. Also use the singlular form fish (not fishes) when referring to a group of individuals of the same species.
Answer: Done.
2. Section 3.2 Second sentence. Macrophages and mast cells are not leukocytes. This sentence should be modified to avoid describing these two cell types as leukocytes.
Answer: This mistake was corrected as follow: “This response has been described in different leukocytes types such as neutrophils [5–7], eosinophils [52], basophils [53], monocytes [54,55], and plasmacytoid dendritic cells [57]. Also, ET formation has been described in macrophages [51] and mast cells [56].” Thank you for this correction.
3. Awkward or long sentences.
-
- Consider re-writing the final sentence in the second last paragraph in Introduction. "Also, endogenous…immune reactions."
Answer: This paragraph war re-wrote as follows: “In line, this pathogen cause endogenous infections of the liver, spleen, serosanguinous ascites and swollen kidneys (nephritis) [33,36,37].”
2. Section 3.2 First paragraph. Consider "...we used a fluorescent dye...to visualize…"
Answer: The paragraph was re-wrote to: “we used a fluorescent dye for extracellular DNA staining (PicoGreen®) to visualize some thin extracellular web-like structures”
3. Section 3.2 Second paragraph. Sentence starting "Nonetheless,..." is difficult to understand.
Answer: This unclear sentence was deleted.
4. Section 3.2 5th paragraph Consider using "The process of ETosis avoids… and hampers..."
Answer: Done.
4. Repeated text: Introduction (second sentence) and Results and Discussion 3.1 (first sentence) are very similar. Alternative formulation should be used to avoid repetition.
Answer: We deleted the repeated text from Results an Discussion. Thank you for this suggestion.
5. Formatting.
-
- Section 2.2 First sentence Degree symbol is underlined
Answer: This mistake was corrected.
2. Section 2.3 Replace comma that is used as decimal point
Answer: This mistake was corrected.
3. Section 3.2 Missing bracket at end of Third sentence
Answer: Done.
6. Spelling.
-
- Abstract. Use image analysis not imaging analysis.
Answer: Done.
2.Section 2.6 In title, replace de with of
Answer: Done.
3. Section 3.2 Second paragraph final sentence. where not were
Answer: This sentence was removed.
4. Section 3.2 Fourth paragraph. remain not remaining
Answer: Done.
7. Word missing.
-
- Introduction. End of second paragraph. "...in the extracellular."
Answer: The sentence was completed to: “entrap and kill microorganisms in the extracellular space”
2. Section 3.2 5th paragraph "...damage of tissue/organs..."
Answer: The sentence was corrected to: “pan histones (H1, H2A/H2B, H3, H4) can damage to tissues or organs neighboring the infection site”
8. Consider using alternative Word.
-
- Introduction. Second last paragraph. "Inline" or rewrite the sentence.
Answer: The sentence was re-wrote to: “Coho salmon (Oncorhynchus kisutch) as well as in rainbow trouts (O. mykiss) showed high mortality rates associated with piscirickettsiosis reaching 14.3% and 46.6%, respectively”.
2. Section 3.1 Second paragraph "Former authors.." Consider e.g. "Samai and coworkers…"
Answer: Done. Thank you for this suggestion.
3. Section 3.2 First paragraph 6th sentence. "Besides" e.g. In addition
Answer: Done.
Reviewer 3 Report
The paper fits the aim of the journal which is very general. The aim of the manuscript, however, it is very specific and it is very well focus on a topic of fish immunology.
The paper is well written and structured. The figures are of good quality and the techniques used are adequate to get the expected results. However, there are two major concerns about this manuscript:
- It is not explained why are included a section of ROS production; this is not explained neither in the abstract nor in the introduction of the paper. If there is an interest in deepen into the relation between ROS production and ETs liberation, the whole manuscript should be adapted to avoid confusions.
- There are no results enough to support a full paper and the discussion is very poor. Additional experiments should be included.
Minor comments: All abbreviations should be described the first time (e.g. PAF).
Author Response
Answers to Reviewers` comments
(Reviewers’ comments are italicized below)
We thank the reviewer by his very useful comments that have greatly improved our paper.
Reviewer 3: Comments and Suggestions for Authors
The paper fits the aim of the journal which is very general. The aim of the manuscript, however, it is very specific and it is very well focus on a topic of fish immunology.
The paper is well written and structured. The figures are of good quality and the techniques used are adequate to get the expected results. However, there are two major concerns about this manuscript:
- It is not explained why are included a section of ROS production; this is not explained neither in the abstract nor in the introduction of the paper. If there is an interest in deepen into the relation between ROS production and ETs liberation, the whole manuscript should be adapted to avoid confusions.
Answer: We added the following sentences:
- “To assess the effect of salmonis on respiratory burst as control of salmon-derived PMN functionality” (in section 2.4. Total reactive oxygen species (ROS) production).
- “and in this work, we used ROS production has functionally validation for isolated salmon-derived PMN”. (in the second paragraph of section: 1. ROS production in P. salmonis-exposed PMN from Atlantic salmon)
2. There are no results enough to support a full paper and the discussion is very poor. Additional experiments should be included.
Answer: We agree that our results are too few to generate a full paper, so the original submission was in the format of “Brief Report” (short communication).
Minor comments: All abbreviations should be described the first time (e.g. PAF).
Answer: Done.
Round 2
Reviewer 3 Report
The authors have improved the manuscript and modfied according to the suggestions given. However, some modifications are still needed and are indicated i nthe attached file
